# The Impact of Soil Water Content on Yield, Composition, Energy, and Water Indicators of the Bioenergy Grass *Saccharum spontaneum* ssp. *aegyptiacum* under Three-Growing Seasons

**Danilo Scordia [1],\*** [ID]**, Silvio Calcagno [1], Alessandra Piccitto [1], Cristina Patanè [2]** [ID] **and Salvatore Luciano Cosentino [1,2]** [ID]

[1] Dipartimento di Agricoltura, Alimentazione e Ambiente (Di3A), University of Catania, via Valdisavoia 5, 95123 Catania, Italy; silvio.calcagno@phd.unict.it (S.C.); alessandra.piccitto@phd.unict.it (A.P.); sl.cosentino@unict.it (S.L.C.)

[2] CNR-Istituto per la BioEconomia (IBE), Sede Secondaria di Catania, Via P. Gaifami 18, 95126 Catania, Italy; cristinamaria.patane@cnr.it

\* Correspondence: dscordia@unict.it; Tel.: +39-095-4783459

**Abstract:** Raising water and energy productivity in agriculture can contribute to reducing the pressure on the limited freshwater availability and non-renewable energy sources. Bioenergy perennial grasses are efficient from a water perspective and can afford a low-energy cultivation system; however, crop selection and cultivation practices for minimizing land use change and maximizing resource use efficiencies remain a challenging task in view of sustainable bioeconomy development. The present work investigated the soil water effect on a long-term plantation of *Saccharum* (*Saccharum spontaneum* ssp. *aegyptiacum*), a bioenergy perennial grass holding great promise for semiarid Mediterranean areas. The plantation was in its 13th year following establishment and was subjected to three levels of irrigation for three successive growing seasons. Regression models between crop water use (CWU) and productivity, biomass composition, energy, and water indicators showed different prediction curves. Raising CWU (from 230 to 920 mm) enhanced the dry biomass yield (from 14.8 to 30.1 Mg ha$^{-1}$) and the net energy value (from 257.6 to 511 GJ ha$^{-1}$). On the same CWU range, unirrigated crops improved the energy efficiency (from 99.8 to 58.5 GJ ha$^{-1}$), the energy productivity (from 5.6 to 3.4 Mg GJ$^{-1}$) and the water productivity (from 114.5 to 56.1 MJ m$^{-3}$) by reducing the water footprint (from 8.7 to 17.8 m$^3$ GJ$^{-1}$). Biomass composition was also superior in unirrigated crops, as the lower heating value, structural polysaccharides, and the acid detergent lignin were higher, while ash and soluble compounds were lower. Present findings demonstrated the good yield levels and persistence of *Saccharum*, improving our knowledge of plant responses to changing soil water availability to maximize energy and conserve natural resources, paving the way for sustainable bioeconomy development in the Mediterranean area.

**Keywords:** dryness; perennial grass; biomass; energy productivity; water footprint; mediterranean

## 1. Introduction

Climate change scenarios predict that extreme weather events will increase in the future. In the Mediterranean area, the frequency of water stress episodes is rising due to prolonged summers and low rainfall, meanwhile, population growth and increasing energy demand is projected to increase water needs between 22% and 74% by the end of the century to stabilize crop production [1–3]. In this context, agriculture needs to secure food, goods, and commodities through adaptation strategies.

On the other hand, agriculture can play a key role to mitigate greenhouse gas emissions (GHG) leading to climate change, and to replace the current fossil fuel based economy with a carbon neutral economy. The transition from a fossil fuel economy to a bioeconomy relies on growing bioenergy crops and on utilizing available crop residues for bioconversion into renewable energies and non-energy bioproducts [4,5].

Prime-quality lands must be committed to sustain food production, thus bioenergy crops should be cultivated on marginal lands to minimize competition for natural resources and changes in land use, and forest and biodiversity losses. Nowadays, there is an increasing interest to define and map marginal lands, and to investigate bioenergy crops suitable for such unproductive areas [5]. In Europe, a considerable loss of utilized agricultural area was recorded in recent decades, mainly due to biophysical constraints to grow food crops, farm structure, and farmer-specific reasons [6]. This risk seems to intensify in the period between 2015 and 2030 [7], and in the Mediterranean substantial areas of agricultural land is already becoming marginal and often abandoned due to increasing dryness [8]. This unprofitable area for food crops could support bioeconomy development through the sustainable intensification of bioenergy crops that use available water efficiently and that are able to grow in low-input cultivation systems [8–10]. Raising water and energy productivity in agriculture, that is, reducing the water footprint and external energy per unit of production, will contribute to reducing the pressure on the limited freshwater resources and non-renewable energy sources [11,12]. The water footprint links the water consumption to the unit biomass production, with a large footprint indicating high demand for water resources while reducing agricultural input such as irrigation, fertilizers, pesticides, and herbicides increases the energy efficiency and reduces the environmental impact associated with the cultivation phase [11–13].

Perennial rhizomatous grasses (PRGs) have demonstrated their capacity to thrive under a range of unfavorable conditions (e.g., drought, salinity, flood and slope), the high energy and water efficiency, and the environmental sustainability [13–17]. Although the energy cost is quite high in the establishment phase, the perennial habit, the lignocellulosic structure, and the capacity to mobilize nutrients makes these species ideal candidates for low-input cultivation systems in the long-term [5]. Among PRGs, $C_4$ photosynthetic plants (i.e., miscanthus and switchgrass) are preferred to the $C_3$ (i.e., reed canarygrass and giant reed) due to a more efficient use of natural resources, such as light, water, and nutrients [18,19].

Apart from the most-investigated PRGs, the Mediterranean is a reach area of nearly unexplored PRGs that are abundantly widespread in hot and drought-prone environments [20,21]. Recently, Cosentino et al. [22] have shown that the PRG *Saccharum spontaneum* ssp. *aegyptiacum*, native to North Africa and naturalized in the south of Italy (the Sicily region), has many traits of a drought-resilient bioenergy crop. It is a $C_4$ plant with high water use efficiency and high biomass yield under both favorable and water stress conditions. Scordia et al. [23] attributed the high yield to a long green leaf area duration, high radiation, and water use efficiency. Under severe water stress, physiological and morphological adjusting mechanisms were detected, suggesting both drought avoidance and tolerance [22,23]. In a four-year field trial in rainfed conditions, this crop showed an average productivity, energy efficiency, and net energy value well above the other native perennial grasses compared side-by-side [21]. Biomass quality improved with the winter harvest as compared with the autumn harvest, as overwinter sustains senescence, the decrease in moisture and ash content, and the increase in structural polysaccharides [21]. *Saccharum* has also shown its potential as a feedstock for biochemical conversion to produce advanced bioethanol [24,25], and key quality traits for low-temperature thermochemical processes [26].

Although *Saccharum* is naturalized in Sicily, it is still an undomesticated crop, and as such needs further insights to optimize the agricultural practices and to understand trade-offs for yield, quality, and resource use efficiencies in a long-term perspective [15,27–29]. The present experiment evaluated the biomass yield and composition (structural polysaccharides, neutral detergent soluble, acid detergent lignin, ash), energy, and water indicators (lower heating value and net energy value, energy efficiency

and productivity, water productivity and footprint) of a long-term plantation of *Saccharum* subjected to changing soil water availability for three successive growing seasons in a semiarid Mediterranean area. Regression models were fitted to understand the rate of change of the dependent variables under a wide range of crop water use given by different irrigation regimes and growing seasons.

## 2. Materials and Methods

### 2.1. Field Trial

The long-term plantation of *Saccharum* (*Saccharum spontaneum* L. ssp. *aegyptiacum* Willd. Hack.) is located at the Experimental farm of the University of Catania, Italy (Figure 1).

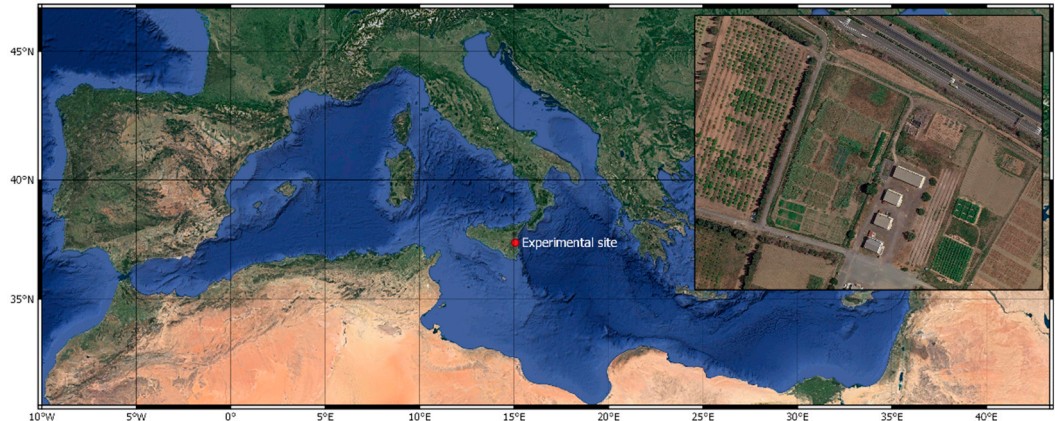

**Figure 1.** Experimental Farm of the University of Catania located in south-eastern Sicily, Italy (10 m a.s.l., 37°24′35.8″ N lat., 15°03′31.7″ E long.).

Details of field trial set-up and crop management were previously reported in Cosentino et al. [22]. Briefly, clonal rhizomes were established in the spring of 2005 in a soil with the following properties: 9.8% silt, 56.6% sand, 33.6% clay, 1.6% organic matter, 0.09 g kg$^{-1}$ total N, 53.4 mg kg$^{-1}$ available P, 300.1 g kg$^{-1}$ exchangeable K, and pH 7.6. The bulk density was 1.1 g cm$^{-3}$. The soil moisture contents at field capacity (at −0.03 MPa) and nominal wilting point (at −1.5 MPa) were 27 g and 11 g H$_2$O 100 g$^{-1}$ dry weight respectively. One rhizome m$^{-2}$ was used in a randomized block design three times replicated with a single plot measuring 15 m$^2$ (5 × 3 m). Before transplanting 100 kg N ha$^{-1}$ and 100 kg P$_2$O$_5$ ha$^{-1}$ as ammonium sulfate and superphosphate, respectively, were supplied. Weeds were controlled manually during the year of establishment. In post-establishment and up to the sixth growing season (2010/2011), neither fertilization nor weed control and pest management were performed. Supplemental irrigation was applied only when leaf rolling was detected, and aboveground biomass was harvested each year during winter-time. From the seventh growing season (2011/12) to the present, a drip irrigation system was used to differentiate the soil water in three levels, namely the 100% of maximum crop evapotranspiration restoration (I$_{100}$–100% ETm), the 50% ETm restoration (I$_{50}$) and no irrigation (I$_0$—rainfed condition). In the same period, crops remained unfertilized, without the necessity to control weeds or pests. The water used for irrigation had a total amount of CaCO$_3$ of 250 ± 24 mg L$^{-1}$, an electrical conductivity of 1300 ± 120 μS cm$^{-1}$ and a pH of 7.5 ± 0.2. The irrigation volume was determined as the maximum available soil water content in a 0.6 m soil depth, and irrigation was scheduled when the sum of daily ETm matched the volume. Rainfall events were subtracted in the daily calculations [22,30].

In the present experiment, irrigation levels described above (I$_{100}$, I$_{50}$ and I$_0$) were compared in three successive growing seasons (2017/18, 2018/19 and 2019/20, hereinafter referred to 2017, 2018 and 2019), corresponding to the 13th, 14th, and 15th yearly harvest, respectively. For each irrigation level,

the crop water use (CWU) was determined by means of water balance from plant re-growth up to the last irrigation:

$$CWU = I + P \pm \Delta C$$

where CWU = crop water use (mm); I = water supplied by means of irrigation (mm); P = precipitation (mm); $\Delta C$ = difference between soil water content at plant re-growth and soil water content after the last irrigation (mm). Soil samples were collected to a depth of 0.6 m in each irrigation treatment, growing season, and replication, at the beginning of each growing season and then when the irrigation was suspended. Fresh soil samples were immediately weighted, closed in plastic zip bags and then transferred in a ventilated oven to dry at 105 °C up to a constant weight.

*2.2. Measurements and Determinations*

Maximum and minimum air temperature and rainfall were measured by a weather station connected to a data logger (Delta-T Devices Ltd, WS-GP1 Compact Weather Station, Cambridge, UK). The reference crop evapotranspiration ($ET_0$) was calculated from the Class A evaporation pan (mm d$^{-1}$) by the pan coefficient of 0.80 [31]. Both equipment were located 150 m from the experimental field. Daily data was aggregated to ten-day increments, from re-sprouting to the end of each growing season.

Aboveground biomass was harvested in winter each year (approx. mid-February) by removing edge plants in all sides of the plots to obtain a sampling area for biomass weight of 6 m$^2$ (3 × 2 m). Biomass was cut 5 cm above ground level and fresh sub-samples were randomly collected, immediately weighted, and then dried to a constant weight at 65 °C. The percentage dry weight was used to calculate the dry biomass yield, which was referred to the unit land area (dry matter yield; DMY, Mg ha$^{-1}$).

Dry sub-samples were then ground through a 1-mm sieve in a mill (IKA-WERFE, Gmbh & Co., KG, Staufenim Breisgau, Germany) for biomass composition determinations. Hemicellulose (HL), cellulose (CL), acid detergent lignin (ADL), ash (ASH) and the neutral detergent soluble (NDS) were determined by a near-infrared spectrometer (NIR, SpectraStar™ 2500XL-R, Unity Scientific, Milford, MA, USA) in a previously developed calibration for lignocellulosic perennial grasses, as reported in Scordia et al. [21]. The biomass lower heating value (LHV) of each sample was obtained by applying the conversion factor proposed by Odum [32] for carbohydrates, proteins, and lipids, modified for structural polysaccharides, ADL and NDS [33].

The LHV (MJ kg$^{-1}$) multiplied by the DMY (Mg ha$^{-1}$) represented the energy output (GJ ha$^{-1}$). As mentioned above, the plantation is managed as low-input; thus weed control, fertilization, and crop protection are not executed. Energy input (GJ ha$^{-1}$) was equivalent to the aboveground biomass harvest and the irrigation. For the harvesting practice (a chipping and loading single-pass system) the same energy consumption was assumed for $I_0$, $I_{50}$, and $I_{100}$ treatments, equivalent to 3.40 GJ ha$^{-1}$ [34]. The energy input for irrigation water of $I_{50}$ and $I_{100}$ treatments was assumed to be 1.0 MJ m$^{-3}$, as reported by Bosatto et al. [35] for the drip irrigation system in the Mediterranean environment.

Energy and water indicators were calculated at the farm gate level, as following: (i) net energy value (NEV, GJ ha$^{-1}$) = energy output − energy input; (ii) energy efficiency (EE, GJ ha$^{-1}$) = energy output/energy input; (iii) energy productivity (EP, Mg GJ$^{-1}$) = dry matter yield/energy input; (iv) water productivity (WP, MJ m$^{-3}$) = energy output/crop water use; (v) water footprint (WF, m$^3$ GJ$^{-1}$) = crop water use/energy output.

*2.3. Statistical Analysis*

Since the same treatments were applied to the same plots for more than one year, collected data were subjected to one-way analysis of variance (ANOVA) using repeated measurements in time, where the growing season represents the within-factor, and the irrigation the between-factor (SPSS, PASW Statistics 18). When data failed Mauchly's sphericity test, the univariate results were adjusted by using the Greenhouse–Geisser Epsilon and the Huynh–Feldt Epsilon correction factors. When univariate results satisfied sphericity tests for within-subjects effects, the *F*-values and associated

*p*-values for between-subjects effects were tested. Differences between means were evaluated for significance using the Student-Newman-Keuls (S.N.K.) test at 95% confidence level.

Relationships between crop water use and measured data were calculated by linear and non-linear regression models. Coefficients were considered significant at $p \leq 0.05$. The Shapiro–Wilk test was developed to test residuals for normality, and the goodness of fit was assessed by calculating $R^2$ (SigmaPlot11, Systat Software Inc., San Jose, CA, USA).

## 3. Results

### 3.1. Meteorological Trend

Air temperature across each growing season was quite similar, ranging from 23.2 to 24.1 °C for the maximum, from 13.0 to 13.4 °C for the minimum, and from 18.3 to 18.7 °C for the mean. It was the lowest in winter, gradually increased from spring to peak in summer and then slowly decreased in autumn down to the lowest values in the next winter (Figure 2).

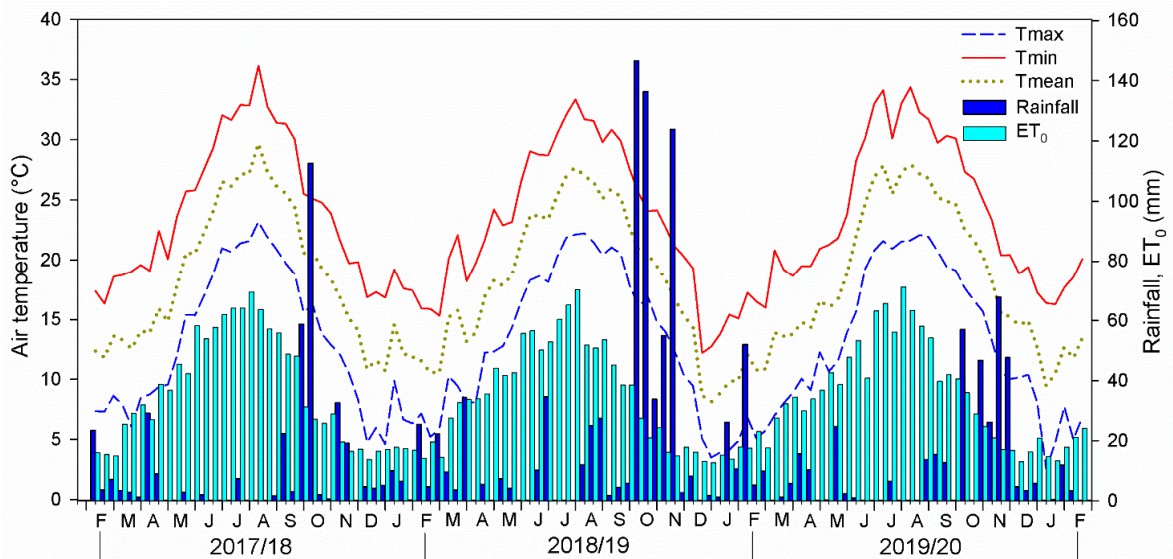

**Figure 2.** Meteorological trend, maximum ($T_{max}$), mean ($T_{mean}$) and minimum ($T_{min}$) air temperatures (°C), rainfall, and reference evapotranspiration ($ET_0$) (mm), at the Experimental Farm of the University of Catania during 2017/18 (2017), 2018/19 (2018) and 2019/20 (2019) growing season.

Rainfall was higher in the 2018 growing season (811 mm) as compared with both 2017 (377 mm) and 2019 (396 mm). The reference evapotranspiration ($ET_0$) was 1290, 1244 and 1299 mm in 2017, 2018, and 2019, respectively. Although rainfall amount was different among growing seasons, distribution patterns narrowly changed among yearly seasons; it was 62–64% in autumn, 9–16% in winter, 8–15% in spring and 9–13% in summer. On the contrary, $ET_0$ was higher in summer (39–41%) than spring (28–32%), autumn (16–17%) and winter (13–14%). The whole season drought index (P/PET) was 0.29, 0.65, and 0.30 in the 2017, 2018, and 2019, respectively. In the period of crop irrigation (from mid-May to the end of August), rainfall was higher in the 2018 (109.2 mm) and $ET_0$ was lower (567.5 mm) as compared with 2017 and 2019 growing seasons (11.0 and 23.2 mm for rainfall and 606.9 and 574.3 mm for ET0, respectively).

The crop water use (CWU) in the 2017, 2018, and 2019 growing season were: 230.1, 357.8, and 259.7 mm in $I_0$; 530.9, 683.6 and 481.9 mm in $I_{50}$; 801.1, 919.4 and 744.3 mm in $I_{100}$, respectively.

### 3.2. Biomass Yield and Composition

ANOVA showed a significant effect of irrigation water (W) and growing season (S) on biomass dry matter yield (DMY), neutral detergent soluble (NDS), hemicellulose (HC), cellulose (CL), acid detergent lignin (ADL), and ash (ASH) content. The W × S interactions were all significant (Table 1).

**Table 1.** Repeated measures analysis of variance (ANOVA) for the main effects and interactions on dry matter yield (DMY), neutral detergent soluble (NDS), hemicellulose (HL), cellulose (CL), acid detergent lignin (ADL), and ash content (ASH) of *Saccharum* under changing irrigation (W) and growing season (S). Degree of freedom (DF); adjusted mean square (Adj MS); Significance indicated by * at $p \leq 0.05$, ** at $p \leq 0.01$, *** at $p \leq 0.001$. LSD value indicates a significant "water × growing season" interaction at $p \leq 0.05$.

| Source | DF | DMY | NDS | HC | CL | ADL | ASH |
|---|---|---|---|---|---|---|---|
| | | **Adj MS** | | | | | |
| Water (W) | 2 | 374.68 *** | 32.22 *** | 3.534 *** | 12.426 *** | 2.959 *** | 2.651 *** |
| Season (S) | 2 | 13.91 *** | 0.443 ** | 0.655 ** | 1.030 ** | 0.146 ** | 0.454 ** |
| W × S | 4 | 4.12 ** | 0.199 ** | 0.120 *** | 0.0196 ** | 0.066 * | 0.0607 ** |
| Error(T) | 12 | 0.095 | 0.009 | 0.001 | 0.001 | 0.003 | 0.003 |
| Error | 6 | 0.433 | 0.109 | 0.005 | 0.004 | 0.006 | 0.016 |
| **W** | **S** | Mg ha$^{-1}$ | % *w/w* | % *w/w* | % *w/w* | % *w/w* | % *w/w* |
| $I_0$ | 2017 | 14.8 ± 0.69 | 18.1 ± 0.37 | 30.4 ± 0.07 | 37.0 ± 0.04 | 8.3 ± 0.03 | 6.1 ± 0.14 |
| $I_{50}$ | 2017 | 25.3 ± 0.28 | 20.8 ± 0.52 | 29.4 ± 0.02 | 35.3 ± 0.03 | 7.7 ± 0.11 | 6.6 ± 0.11 |
| $I_{100}$ | 2017 | 29.3 ± 0.42 | 22.5 ± 0.12 | 28.8 ± 0.05 | 34.2 ± 0.02 | 6.9 ± 0.04 | 7.4 ± 0.06 |
| $I_0$ | 2018 | 19.2 ± 0.40 | 19.0 ± 0.39 | 29.5 ± 0.07 | 35.8 ± 0.08 | 7.9 ± 0.06 | 6.7 ± 0.11 |
| $I_{50}$ | 2018 | 27.3 ± 0.41 | 21.0 ± 0.14 | 29.0 ± 0.12 | 34.7 ± 0.09 | 7.3 ± 0.06 | 7.2 ± 0.18 |
| $I_{100}$ | 2018 | 30.1 ± 0.25 | 22.7 ± 0.09 | 28.5 ± 0.07 | 34.1 ± 0.07 | 7.0 ± 0.11 | 7.6 ± 0.5 |
| $I_0$ | 2019 | 16.8 ± 1.01 | 18.5 ± 0.71 | 29.8 ± 0.01 | 36.7 ± 0.06 | 8.2 ± 0.04 | 6.3 ± 0.10 |
| $I_{50}$ | 2019 | 24.2 ± 0.30 | 20.7 ± 0.15 | 28.9 ± 0.14 | 35.3 ± 0.02 | 7.6 ± 0.07 | 6.9 ± 0.20 |
| $I_{100}$ | 2019 | 29.3 ± 1.83 | 22.5 ± 0.11 | 28.6 ± 0.06 | 34.2 ± 0.10 | 6.9 ± 0.12 | 7.3 ± 0.07 |
| LSD$_{(W \times S)}$ | ≤0.05 | 1.044 | 0.524 | 0.112 | 0.100 | 0.123 | 0.634 |

Generally, the wetter the growing season (as in 2018) or the greater the CWU ($I_{100} > I_{50} > I_0$), the higher the DMY, the NDS, and the ASH content. Drier seasons (as in 2017 and 2019) or lower CWU improved the HL, the CL, and the ADL content. Overall, the significantly highest DMY was registered in $I_{100}$ in 2018 (30.1 Mg DM ha$^{-1}$), while the significantly lowest was in $I_0$ in the 2017 growing season (14.8 Mg DM ha$^{-1}$). Similar patterns were observed for NDS (22.7% in $I_{100}$ in 2018 and 18.1% in $I_0$ in 2017) and ASH content (7.6% in I100 in 2018 and 6.1% in $I_0$ in 2017).

On the contrary, HL, CL, and ADL content were the highest in $I_0$ in 2017 (30.4%, 30.7%, and 8.3%, respectively), and the lowest in $I_{100}$ in 2018 (28.5%, 34.1%, 7.0%, respectively). $I_{50}$ treatment was at the middle range of statistical significance for both biomass yield and composition.

### 3.3. Energy and Water Indicators

ANOVA showed a significant effect of irrigation water (W), growing season (S), and the W × S interactions on lower heating value (LHV), net energy value (NEV), energy efficiency (EE), energy and water productivity (EP and WP, respectively), and water footprint (WF) (Table 2).

Generally, the wetter the growing season (as in 2018) or the greater the CWU ($I_{100} > I_{50} > I_0$), the higher the NEV and the WF. Drier seasons (as in 2017 and 2019) or lower CWU improved the LHV, EE, EP, and WP. Overall, the highest and the lowest LHV were registered in $I_0$ in 2017 (17.7 MJ kg$^{-1}$) and in $I_{100}$ in 2018 (17.1 MJ kg$^{-1}$), respectively. $I_0$ also showed the highest EE and EP in 2018 (99.8 GJ ha$^{-1}$ and 5.56 Mg GJ$^{-1}$, respectively), and the highest WP (114.5 MJ m$^{-3}$) in 2019.

**Table 2.** Repeated measures ANOVA for main effects and interactions on net energy value (NEV), energy efficiency (EE), energy productivity (EP), water productivity (WP), and water footprint (WF) of *Saccharum* under changing irrigation (W) and growing season (S). Degree of freedom (DF); adjusted mean square (Adj MS); Significance indicated by * at $p \leq 0.05$, ** at $p \leq 0.01$, *** at $p \leq 0.001$. LSD value indicates a significant "irrigation × growing season" interaction at $p \leq 0.05$.

| Source | DF | LHV | NEV | EE | EP | WP | WF |
|---|---|---|---|---|---|---|---|
| | | | | **Adj MS** | | | |
| Water (W) | 2 | 0.3742 *** | 106412 *** | 1377.15 *** | 3.857 *** | 4644.3 *** | 102.62 *** |
| Season (S) | 2 | 0.0050 * | 3633 *** | 539.70 *** | 1.975 *** | 715.9 *** | 17.66 *** |
| W × S | 4 | 0.0025 * | 1822 ** | 75.52 ** | 0.161 * | 31.85 *** | 0.405 * |
| Error(T) | 12 | 0.0001 | 21.0 | 0.62 | 0.002 | 2.13 | 0.068 |
| Error | 6 | 0.0007 | 119.0 | 5.48 | 0.018 | 54.98 | 0.856 |
| W | S | MJ kg$^{-1}$ | GJ ha$^{-1}$ | GJ ha$^{-1}$ | Mg GJ$^{-1}$ | MJ m$^{-3}$ | m$^3$ GJ$^{-1}$ |
| $I_0$ | 2017 | 17.7 ± 0.01 | 294.9 ± 12.2 | 87.8 ± 3.59 | 4.35 ± 0.20 | 113.7 ± 6.97 | 8.79 ± 0.70 |
| $I_{50}$ | 2017 | 17.5 ± 0.04 | 437.5 ± 3.93 | 73.6 ± 0.61 | 4.20 ± 0.04 | 83.5 ± 1.35 | 12.0 ± 0.14 |
| $I_{100}$ | 2017 | 17.2 ± 0.02 | 497.4 ± 7.87 | 58.5 ± 0.83 | 3.39 ± 0.05 | 63.2 ± 3.01 | 15.8 ± 0.31 |
| $I_0$ | 2018 | 17.5 ± 0.02 | 336.2 ± 6.81 | 99.8 ± 2.01 | 5.65 ± 0.12 | 94.1 ± 4.86 | 10.6 ± 0.49 |
| $I_{50}$ | 2018 | 17.3 ± 0.03 | 471.7 ± 6.02 | 88.1 ± 1.05 | 5.04 ± 0.07 | 69.4 ± 2.75 | 14.4 ± 0.27 |
| $I_{100}$ | 2018 | 17.1 ± 0.01 | 511.0 ± 4.95 | 69.7 ± 0.69 | 4.04 ± 0.03 | 56.1 ± 4.86 | 17.8 ± 0.05 |
| $I_0$ | 2019 | 17.6 ± 0.03 | 257.6 ± 17.5 | 76.8 ± 5.15 | 4.96 ± 0.29 | 114.5 ± 7.76 | 8.73 ± 0.77 |
| $I_{50}$ | 2019 | 17.4 ± 0.04 | 416.0 ± 3.88 | 72.4 ± 0.64 | 4.15 ± 0.05 | 87.5 ± 1.45 | 11.4 ± 0.15 |
| $I_{100}$ | 2019 | 17.2 ± 0.01 | 519.1 ± 32.2 | 63.9 ± 3.72 | 3.71 ± 0.21 | 67.9 ± 1.10 | 14.7 ± 1.10 |
| LSD$_{(W × S)}$ | ≤0.05 | 0.043 | 17.30 | 3.714 | 0.215 | 11.75 | 1.468 |

The highest overall NEV and WF were associated to $I_{100}$ in 2018 (511 GJ ha$^{-1}$ and 17.8 m$^3$ GJ$^{-1}$, respectively), and the lowest to $I_0$ in 2019 (257.6 GJ ha$^{-1}$ and 8.7 m$^3$ GJ$^{-1}$, respectively). Even in this case, the $I_{50}$ treatment was at the middle range for both energy and water indicators.

### 3.4. Relationships among Crop Water Use, Yield, Composition, Energy and Water

The fitted equations between crop water use (CWU) and biomass yield (DMY), and CWU and biomass composition (NDS, HL, CL, ADL and ASH) had significant estimated parameters and small standard errors of the regressions contributing to a good adaption of fitted data (Table 3).

**Table 3.** Estimated coefficients, standard error, *t*-and *p*-value of the fitted equations between crop water use and biomass yield and composition of *Saccharum* under changing irrigation and growing season.

| Relationship | Equation | Coefficient | Value | SE | *t*-Value | *p*-Value |
|---|---|---|---|---|---|---|
| CWU vs. DMY | $y = a \times (1 - \exp^{(-bx)})$ | a | 33.635 | 0.774 | 43.430 | <0.001 |
| | | b | 0.0024 | 0.0001 | 18.759 | <0.001 |
| CWU vs. NDS | $y = y0 + ax$ | y0 | 16.772 | 0.3814 | 43.976 | <0.001 |
| | | a | 0.007 | 0.0006 | 11.038 | <0.001 |
| CWU vs. HC | $y = y0 + \left(\frac{a}{x}\right)$ | y0 | 28.153 | 0.1657 | 169.888 | <0.001 |
| | | a | 485.588 | 66.400 | 7.313 | 0.002 |
| CWU vs. CL | $y = y0 + \left(\frac{a}{x}\right)$ | y0 | 33.261 | 0.1662 | 200.106 | <0.001 |
| | | a | 892.410 | 66.601 | 13.399 | <0.001 |
| CWU vs. ADL | $y = y0 + \left(\frac{a}{x}\right)$ | y0 | 6.622 | 0.1381 | 47.939 | <0.001 |
| | | a | 419.799 | 55.351 | 7.584 | 0.001 |
| CWU vs. ASH | $y = y0 + ax$ | y0 | 5.7670 | 0.1183 | 48.7447 | <0.001 |
| | | a | 0.0021 | 0.0002 | 10.5519 | <0.001 |

CWU (crop water use, mm); DMY (dry matter yield, Mg ha$^{-1}$); NDS (neutral detergent soluble, % *w/w*); HC (hemicellulose, % *w/w*); CL (cellulose, % *w/w*); ADL (acid detergent lignin, % *w/w*); ASH (ash, % *w/w*).

The normality test indicated that residuals were normally distributed, and regressions showed high goodness of fit ($R^2$ from 0.88 to 0.99; Figure 3). The relationship between CWU and DMY was described by an asymptotic equation, with maximum DMY value of 33.63 Mg ha$^{-1}$. Within the range of the observed data the DMY increased almost linearly, from 14.9 to 20.2 Mg ha$^{-1}$ at CWU in the range of 230–360 mm (corresponding to the observed data of $I_0$ treatment). At higher CWU (450–700 mm), approximately in the range of the $I_{50}$ treatment, the DMY increase was less than proportional (from 23.1 to 28.0 Mg ha$^{-1}$). Further increase in the CWU (750–920 mm) led to small increases of DMY (from 28.7 to 30.4 Mg ha$^{-1}$, corresponding to the $I_{100}$ treatment). The relationships between CWU and NDS, and between CWU and ASH, were described by linear equations, indicating that within the limit of the observed data both depended variables increased linearly with the CWU. Hence, the lowest NDS and ASH values were in $I_0$ and proportionally increased in $I_{50}$ and in $I_{100}$ (from 18.0% at CWU of 230 mm to 23.2% at CWU of 920 mm for NDS, and from 6.2 to 7.7% for ASH in the same CWU range). The relationships between CWU and structural polysaccharides (HL and CL, respectively), and CWU and ADL were described by inverse polynomial equations. The trends showed a rapid decrease of the dependent variables as the CWU increased from 230 to 360 mm (from 30.3% to 29.5% for HL, from 37.1% to 35.7% for CL and from 8.5% to 7.8% for ADL), followed by a lower rate of change at CWU of 450–680 mm (from 29.2% to 28.9% for HL, from 35.8% to 34.6% for CL and from 7.6% to 7.2% for ADL) to an almost flat rate at CWU of 700–920 mm (from 28.8% to 28.7% for HL, from 34.5% to 34.2% for CL and from 7.2% to 7.1% for ADL).

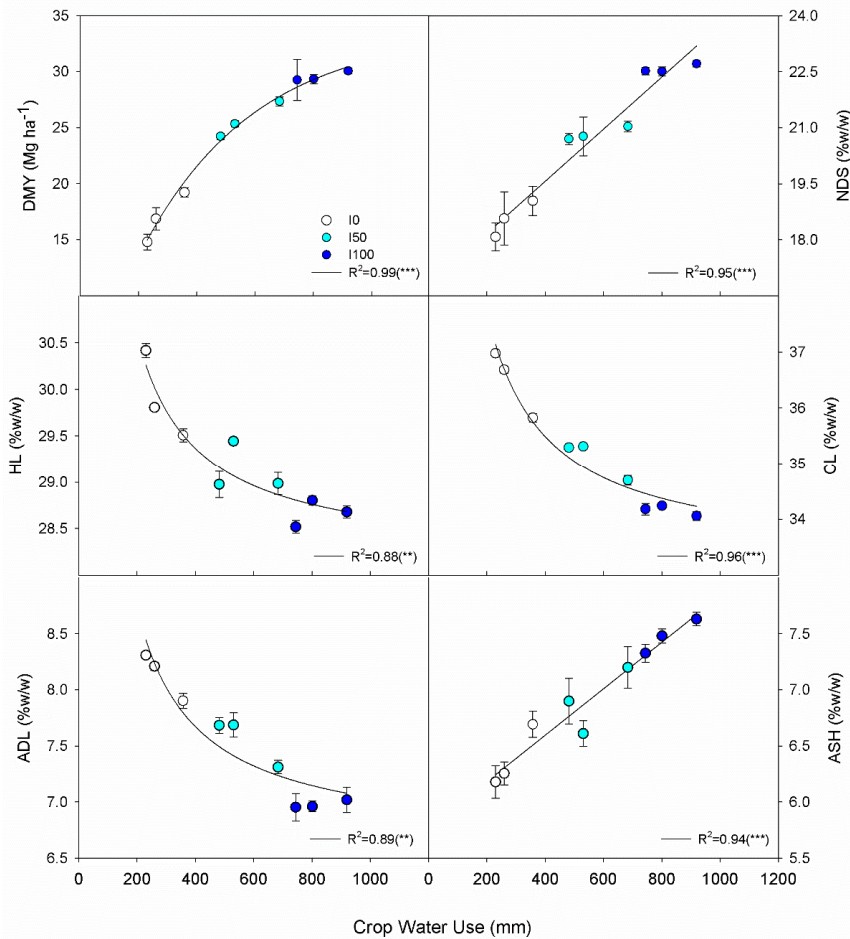

**Figure 3.** Relationships between crop water use (mm) and dry matter yield (DMY), and biomass composition (neutral detergent soluble—NDS, hemicellulose—HL, cellulose—CL, acid detergent lignin—ADL, and ash content) of *Saccharum* under changing irrigation and growing season. Regression significance indicated by ** at $p \leq 0.01$, *** at $p \leq 0.001$.

The fitted equations between crop water use (CWU) and energy indicators (LHV, NEV, EE, and EP), and between CWU and water indicators (WP and WF) showed high significance of the estimated parameters, except for the relationships between CWU and EE, and CWU and EP, where "*a*" coefficients, which represent the slopes of the linear equations, were not significant (Table 4).

**Table 4.** Estimated coefficients, standard error, *t*- and *p*-value of the fitted equations between crop water use and energy and water indicators of *Saccharum* under changing irrigation and growing season.

| Relationship | Equation | Coefficient | Value | SE | *t*-Value | *p*-Value |
|---|---|---|---|---|---|---|
| CWU vs. LHV | $y = y_0 + ax$ | $y_0$ | 17.8392 | 0.0360 | 495.4335 | <0.001 |
| | | a | −0.0007 | 0.00005 | −12.5399 | <0.001 |
| CWU vs. NEV | $y = a \times (1 - \exp^{(-bx)})$ | a | 570.603 | 26.850 | 21.251 | <0.001 |
| | | b | 0.0027 | 0.0003 | 8.905 | <0.001 |
| CWU vs. EE | $y = y_0 + ax$ | $y_0$ | 94.9085 | 9.5250 | 9.9642 | <0.001 |
| | | a | −0.0326 | 0.0158 | −2.0648 | 0.078 |
| CWU vs. EP | $y = y_0 + ax$ | $y_0$ | 5.3091 | 0.5375 | 9.8771 | <0.001 |
| | | a | −0.0016 | 0.0009 | −1.8482 | 0.107 |
| CWU vs. WP | $y = y_0 + ax$ | $y_0$ | 129.8059 | 3.8662 | 33.5744 | <0.001 |
| | | a | −0.0829 | 0.0064 | −12.9270 | <0.001 |
| CWU vs. WF | $y = y_0 + ax$ | $y_0$ | 5.8007 | 0.2569 | 22.5805 | <0.001 |
| | | a | 0.0122 | 0.0004 | 28.7397 | <0.001 |

CWU (crop water use, mm); LHV (lower heating value, MJ $kg^{-1}$); NEV (net energy value GJ $ha^{-1}$); EE (energy efficiency, GJ $ha^{-1}$); EP (energy productivity, Mg $GJ^{-1}$); WP (water productivity, MJ $m^{-3}$); WF (water footprint ($m^3$ $GJ^{-1}$).

With the exception of CWU and EE, and CWU and EP, regressions were significant and showed high goodness of fit ($R^2$ ranging from 0.94 to 0.99; Figure 4). The relationship between CWU and LHV was predicted by a linear, negative trend, indicating a steady decrease of LHV as the CWU increased (from 17.7 MJ $kg^{-1}$ at 230 mm to 17.1 MJ $kg^{-1}$ at 920 mm of CWU). Although regressions were not significant, negative linear trends were also predicted for the CWU and EE (from 87.4 GJ $ha^{-1}$ at 230 mm to 64.9 GJ $ha^{-1}$ at 920 mm of CWU), and for the CWU and EP (from 4.9 Mg $GJ^{-1}$ at 230 mm to 3.8 Mg $GJ^{-1}$ at 920 mm of CWU). A negative, linear trend was also found for the relationships between CWU and WP, indicating a proportional decrease of WP with CWU (from 110.7 to 53.6 MJ $m^{-3}$ at CWU from 230 to 920 mm).

The asymptotic equation, which described the CWU and NEV relationship, predicted a maximum NEV value of 570.6 GJ $ha^{-1}$ at the highest CWU level. NEV increased almost linearly at CWU in the range of 230–360 mm (from 262.0 to 352.1 GJ $ha^{-1}$), followed by a less than proportional increasing trend at CWU from 450 to 700 mm (from 399.5 to 482.9 GJ $ha^{-1}$), and a further lower rate of change at CWU between 750 and 920 mm (from 493.6 to 521.6 GJ $ha^{-1}$).

On the other hand, the relationship between CWU and WF was described by a positive linear equation, indicating that the WF proportionally increased with CWU (from 8.6 $m^3$ $GJ^{-1}$ at 230 mm to 17.1 $m^3$ $GJ^{-1}$ at 920 mm).

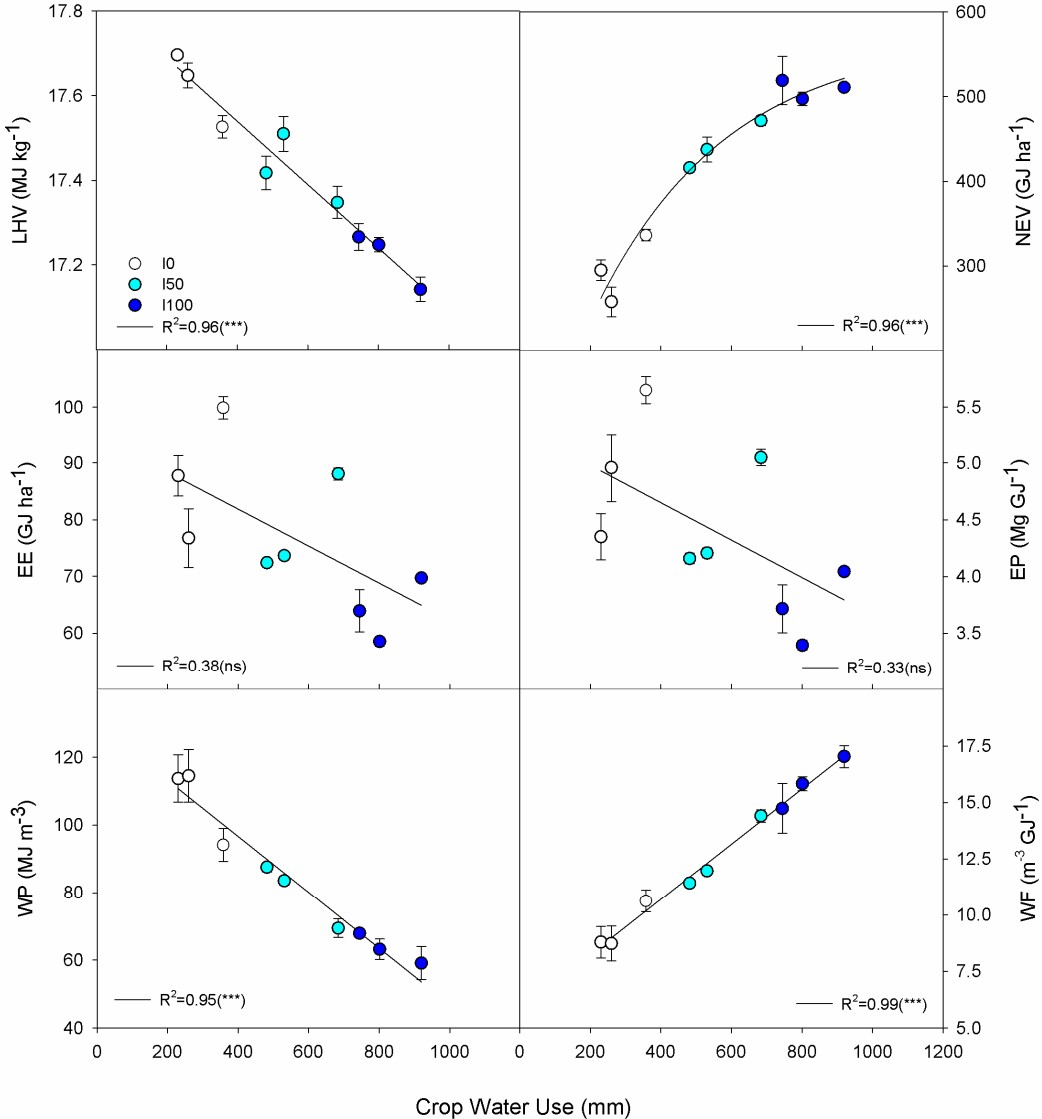

**Figure 4.** Relationships between crop water use (mm) and energy indicators (lower heating value—LHV, net energy value—NEV, energy efficiency—EE, energy productivity—EP) and water indicators (water productivity—WP, water footprint—WF) of *Saccharum* under changing irrigation and growing season. Regression significance indicated by *** at $p \leq 0.001$, or (ns) not significant.

## 4. Discussion

Global demands for food, water, and energy will rise over the next decades [36], posing further challenges to the Mediterranean agriculture that already faces bio-physical and socio-economic limitations [3,37]. It is imperative that agriculture will take these threats as opportunities, by diversifying crop options, introducing new cropping systems with locally adapted genotypes combined with sustainable agronomic practices to provide raw material for the bioeconomy and to conserve natural resources [38]. Advanced research on locally adapted plants, aimed at identifying stress-resilient traits and to breed resource-efficient and climate-resilient genotypes, will likely play a key role in strength synergies among farmers, practitioners, and industries to optimize bioeconomy value chains [39–44].

The present study improved our knowledge of the long-term behavior of the perennial bioenergy grass *Saccharum*, which is naturalized in the south Mediterranean, providing reliable information on biomass yield and composition, energy, and water indicators, which are fundamental to sustainably intensify perennial cropping systems in this environment and in similar semiarid conditions worldwide. The meteorological parameters throughout the experimental period highlighted a trend towards drier

summer seasons, with high risk of prolonged drought and substantial inter-annual precipitation variability. Two out the three growing seasons (2017 and 2019) had a dryness index (P/PET) well below the threshold of 0.5 set in the study of the Joint Research Center for delineation of drought affected areas [45]. Although the 2018 was a wet season, the largest part of precipitation (around 500 mm) were registered in the fall when plants are at the onset of senescence and water uptake is limited [23]. In the spring-summer period, corresponding to the rapid vegetative crop development, the $ET_0$ was five-fold higher than precipitation in 2018. The $ET_0$ was eleven-fold and eight-fold higher than precipitation in the spring–summer of 2017 and 2019, respectively, exacerbating drought conditions in this crucial phase of the crop's biological cycle [46,47].

Nonetheless, *Saccharum* demonstrated good yield levels and persistence of the plantation, even under rainfed conditions and variable growing seasons, suggesting it possesses drought-resilient traits to cope with water stress episodes and prolonged summer droughts [48,49]. As expected, raising soil water content (CWU) by means of irrigation significantly enhanced the biomass yield (DMY) and the net energy value (NEV). On the contrary, rainfed crops significantly improved the energy efficiency (EE), the energy productivity (EP) and the water productivity (WP) by raising the water footprint (WF) per unit of production. Biomass composition was also superior in rainfed than irrigated crops, as the lower heating value (LHV), structural polysaccharides (hemicellulose—HL and cellulose—CL) and acid detergent lignin (ADL) were significantly higher, while ash (ASH) and soluble content (NDS) were significantly lower. This is of utmost importance for thermochemical processes [26], but is less influential for biochemical conversions since an increase in structural polysaccharides was accompanied by an increase in ADL, which is notoriously a barrier to hydrolytic enzymes [50].

Fitting models predicted most of the relationships analyzed quite well, helping to answer all of the research questions underpinning the problem posed in this study.

The DMY as function of CWU was described by an asymptotic equation, in accordance with our previous study on *Saccharum* at an early stage of its lifespan, namely in the 7–9th year following the establishment [22]. While the slope values were similar, the DMY asymptotic value was higher in the previous investigation (37.86 vs. 33.63 Mg ha$^{-1}$), which agrees with the long-term behavior of perennial rhizomatous grasses. Indeed, depending on environmental and growing conditions, many authors argued for an upward DMY trend in the first 4 to 8 years, followed by a declining phase after 10–12 years [15,51–53]. The present results refer to a plantation in the 13–15th year, therefore in the second phase of the plantation lifespan. The lowest DMY found in rainfed crops ($I_0$), particularly when the driest growing season occurred in 2017 (DMY was 56% lower the asymptotic value), corroborated the importance of precipitation amount and distribution for unirrigated crops in this area [15,21,33].

The NEV, which expresses the gained difference in energy between the energy content of the biomass and the energy expended to produce it, showed a similar asymptotic trend found for the DMY. This NEV range is slightly higher than *Miscanthus* × *giganteus* and slightly lower than *Arundo donax* cultivated in the South Mediterranean in the first four years following the establishment [54]. In rainfed conditions, the present NEV averaged 295.3 GJ ha$^{-1}$ yr$^{-1}$ over three years, while a rainfed *Saccharum* plantation had a NEV of ~335 GJ ha$^{-1}$ yr$^{-1}$ as the average of the first three years following the establishment [21], which agrees with the long-term behavior mentioned above.

The increasing, linear trend predicted for the NDS and the ASH with CWU suggested that irrigated crops ($I_{100}$ and $I_{50}$) had a higher availability of non-structural components and minerals in their cell walls than rainfed crops. This could be explained by the higher ratio of green leaves to stems at harvest in irrigated crops (data not shown), as leaves in grasses generally contain more ash but also more protein, soluble carbohydrates, and other non-structural components [55,56].

On the other hand, structural compound of the plant cell wall (i.e., HL, CL and ADL) to the CWU were predicted by inverse polynomial equations, with the largest value at the lowest CWU levels; the rate of change of the depended variables was rapid at low CWU values, reduced at intermediate CWU and leveled off at the highest CWU. Apart from the water effect, the higher stem to leaf ratio in rainfed crops likely improved structural polysaccharides. A similar

downward trend was previously observed in the study by Cosentino et al. [22], where rainfed *Saccharum* crops showed higher hemicellulose, cellulose, and lignin than mid and full irrigated crops. Biomass composition in *Saccharum* was comparable with levels found for many other perennial grasses, such as *Miscanthus × giganteus*, *Miscanthus sinensis*, *Arundo donax*, *Cymbopogon hyrthus*, *Sorghum halepense*, *Oryzopsis milianea*, and *Panicum virgatum* [21,25,26,56,57].

The negative, linear relationship between the LHV and CWU corroborates the negative correlation between LHV and ash, and the positive correlation between LHV and structural polysaccharides and lignin content [26,58]. Similar observations were already reported in the younger *Saccharum* stand, where the $I_0$ had the significantly highest LHV, $I_{100}$ the lowest and $I_{50}$ was at the middle range [22].

The EE and the EP were also negatively related with the CWU. Although both regressions were not significant and the goodness of fit were quite low ($R^2$ of 0.38 and 0.33, respectively), the decreasing trend suggested that both energy indicators were depended on the energy expended to produce one unit of output. It follows that the biomass energy content over the energy invested per unit land (EE) and the biomass produced per unit energy invested (EP), respectively, were the highest when the irrigation water was not provided, as the energy cost for the drip irrigation was assumed at 1 MJ per $m^3$ of water supplied [35]. A rainfed *Saccharum* cultivated in the same area and similar agricultural management showed an average EE of around 110 GJ ha$^{-1}$ yr$^{-1}$, with a peak of 154.8 GJ ha$^{-1}$ at the third harvest [21]. In the present study, the three-year average EE was 88.1 GJ ha$^{-1}$ yr$^{-1}$ under rainfed conditions, highlighting that long-term energy efficiency and productivity was supported by a good persistence of the stand. When irrigated at full water restoration ($I_{100}$), the EE in the three-growing seasons reduced from 16% to 33% and the EP from 22% to 28%.

The negative relationship between the WP and the CWU showed that the biomass energy content associated with a one-unit shift in the water supplied decreased linearly (from 114.5 MJ m$^{-3}$ in $I_0$ to 56.1 MJ m$^{-3}$ in $I_{100}$), which agrees with the relationships found between the water use efficiency (WUE, g L$^{-1}$) and the CWU of *Miscanthus × giganteus* [59], *Arundo donax* [30], and *Saccharum* [22] tested in the same location. Such WP corresponds to a WUE of 6.0 in $I_0$, 4.6 in $I_{50}$ and 3.6 g L$^{-1}$ yr$^{-1}$ in $I_{100}$ across the three growing seasons, which is quite comparable to the WUE values of 5.9 ($I_0$), 4.2 ($I_{50}$), and 3.3 g L$^{-1}$ ($I_{100}$) found by Cosentino et al. [22] with this *Saccharum* plantation in the 2013 growing season (the 9th year following the establishment).

The linear increase of WF raising the CWU confirmed that rainfed crops improved the cultivation system with regard to the volume of water consumption per GJ of biomass produced. Averaging over the three-growing seasons, the WF in $I_0$ was 9.4 m$^3$ GJ$^{-1}$ yr$^{-1}$, increasing by 34% in $I_{50}$ and by 72% in $I_{100}$. It has been shown that the WF changes with crops, cultivation systems, and environmental conditions [12]. The global average water footprint per ton of crop has been estimated to be 200 m$^3$ Mg$^{-1}$ in sugar crops, 1000 m$^3$ Mg$^{-1}$ in fruit, 1600 m$^3$ Mg$^{-1}$ in cereals, and 2400 m$^3$ Mg$^{-1}$ in oil crops [60]. The present results converted into dry tons had a lower WF than even sugar crops when *Saccharum* used only precipitations (from 154.1 to 186.2 m$^3$ Mg$^{-1}$ in $I_0$). However, the WF was larger when the irrigation was provided to the *Saccharum* plantation (from 199.0 to 249.9 m$^3$ Mg$^{-1}$ in $I_{50}$ and from 254.4 to 305.7 m$^3$ Mg$^{-1}$ in $I_{100}$), suggesting the importance of water management in Mediterranean cropping systems. The WF of the perennial bioenergy grass *Miscanthus* was 334, 629, 828, and 1082 m$^3$ Mg$^{-1}$ in different environmental conditions, namely those found in North Europe, United States, Brazil, and Zimbabwe, respectively [61]. The present long-term plantation of *Saccharum* showed a reduced WF over the best results achieved by *Miscanthus* in the North Europe, indicating the great potential of locally adapted species in the sustainable use of natural resources.

## 5. Conclusions

Successful cultivation of bioenergy crops in drought affected areas requires a deep understanding of the interaction between genotype and environment across successive growing seasons, particularly when new cropping systems are employed and climatic conditions are difficult to predict. Locally adapted genotypes, such as *Saccharum spontaneum* ssp. *aegyptiacum*, which is naturalized in

southern Italy, demonstrate good yield levels and persistence of the plantation despite a meteorological trend towards drier summer seasons and strong inter-annual precipitation variability.

Biomass dry matter yield and net energy value can be maximized by using irrigation water. On the other hand, cultivating *Saccharum* without irrigation improved the biomass composition, the energy efficiency, the energy productivity, and the water productivity by reducing the water footprint.

The main findings suggest a high energy sustainability of this crop grown in the absence of input (irrigation, fertilization, crop protection) for more than ten years, which contributes to enhancing our knowledge of sustainable crop management across the south Mediterranean area and in similar semiarid environmental conditions worldwide.

Using drought-resilient species and tailored agronomic practice represents a worthwhile strategy towards approaches that are more sustainable and resilient against future challenges. It is worth mentioning that *Saccharum* is still an undomesticated wild-type species with great genetic potential for improvements. Further research on phenotypic selections of outstanding genotypes from diverse environments across the Mediterranean basin would potentially allow the selection of key traits for improving yield, quality, and drought-resilience. Finally, an integrated environmental assessment, including a soil nitrogen balance, would provide a more holistic approach to the long-term sustainability of the system as a whole. However, as with similar perennial bioenergy grasses, advanced research in several areas is still required (large-scale cultivation, non-invasiveness, maintenance of biodiversity, ecosystem services, etc.) in order to implement effective supply chains from the field to the biorefinery based on *Saccharum* as a raw material.

**Author Contributions:** Conceptualization by D.S.; Data curation by D.S.; Formal analysis by D.S.; Funding acquisition by S.L.C.; Investigation by D.S., S.C. and A.P.; Methodology by D.S.; Resources by S.L.C.; Software by D.S.; Supervision by D.S.; Validation by D.S.; Visualization by D.S.; Writing—original draft by D.S.; Writing—review & editing by D.S., C.P. and S.L.C. All authors have read and agreed to the published version of the manuscript.

**Funding:** This paper is part of a project that has received funding from the European Union's Horizon 2020 research and innovation programme under grant agreement No 727698, project MAGIC (Marginal lands for growing industrial crops: turning a burden into an opportunity).

**Acknowledgments:** Authors gratefully acknowledge Matteo Maugeri, Santo Virgillito and Giancarlo Patanè of the "Dipartimento di Agricoltura, Alimentazione e Ambiente (Di3A), University of Catania" for field trial set-up and maintenance.

**Conflicts of Interest:** The authors declare no conflict of interest.

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
