# Peer review of "The Impact of Soil Water Content on Yield, Composition, Energy, and Water Indicators of the Bioenergy Grass Saccharum spontaneum ssp. aegyptiacum under Three-Growing Seasons"

_agronomy, doi:10.3390/agronomy10081105_

Round 1

Reviewer 1 Report

Dear authors,

because this is your new resubmission, I have carefully checked all your answers and correction incorporated into the original manuscript. I was happy with all the corrections and I have found this paper more suitable for the readers.

I strongy recommend to add the map of the sampling locality.

Congratulations and good luck with your future research.

Author Response

Reviewer 1

Dear authors,

because this is your new resubmission, I have carefully checked all your answers and correction incorporated into the original manuscript. I was happy with all the corrections and I have found this paper more suitable for the readers.

Dear reviewer #1, we really thank you for providing valuable comments to improve our manuscript. We greatly appreciate the time taken by yourself to assess our contribution. As suggested, we have spell checked thoroughly the text, tables and figures for English language and style.

I strongly recommend to add the map of the sampling locality.

Following your recommendation, as well as the comment raised by reviewer #3, we have provided a map of the experimental site where the trial was carried out (please see figure 1 of the revised version).

Congratulations and good luck with your future research.

Thank you very much.

Reviewer 2 Report

Dear Authors,

I congratulate to you on such a hard and devoted work and finally, your success. Thank you for your kind answers to my suggestions and comments. Wish you and your team all the best and good luck in future scientific work.  

Author Response

Reviewer 2

Dear Authors,

I congratulate to you on such a hard and devoted work and finally, your success. Thank you for your kind answers to my suggestions and comments. Wish you and your team all the best and good luck in future scientific work.  

Dear reviewer #2, we really thank you for providing valuable comments to improve our manuscript. We greatly appreciate the time taken by yourself to assess our contribution.

Reviewer 3 Report

Thank you for the thorough consideration of my comments, and the excellent additions to the manuscript.

For my comment in the "Materials and Methods" section, sorry I provided you with a wrong link.

Here is the correct link (//tiny.cc/30zjrz). Here I recommended you adding a map for the study area; including the below properties.
Projected Coordinate System: WGS_1984_UTM_Zone_21N
Projection: Transverse Mercator
Linear Unit: Meter
Geographic Coordinate System: GCS_WGS_1984
Datum: D_WGS_1984
Prime Meridian: Greenwich
Angular Unit: Degree

Author Response

Reviewer 3

Thank you for the thorough consideration of my comments, and the excellent additions to the manuscript.

Dear reviewer #3, we really thank you for providing valuable comments to improve our manuscript. We greatly appreciate the time taken by yourself to assess our contribution. As suggested, we have spell checked thoroughly the text, tables and figures for English language and style.

For my comment in the "Materials and Methods" section, sorry I provided you with a wrong link.

Here is the correct link (//tiny.cc/30zjrz). Here I recommended you adding a map for the study area; including the below properties. 
Projected Coordinate System: WGS_1984_UTM_Zone_21N
Projection: Transverse Mercator
Linear Unit: Meter
Geographic Coordinate System: GCS_WGS_1984
Datum: D_WGS_1984
Prime Meridian: Greenwich
Angular Unit: Degree

Thank you for providing the correct link. Following your recommendation, we have provided a map of the experimental site where the trial was carried out (please see figure 1 of the revised version).
